# Examining the Association Between Overweight, Obesity, and Irritable Bowel Syndrome: A Systematic Review and Meta-Analysis

**DOI:** 10.3390/nu16233984

**Published:** 2024-11-21

**Authors:** Chun En Yau, Gwendolyn Shan Jing Lim, Asher Yu Han Ang, Yu Liang Lim, Orlanda Qi Mei Goh, Kewin Tien Ho Siah, Qin Xiang Ng

**Affiliations:** 1NUS Yong Loo Lin School of Medicine, National University of Singapore, Singapore 117597, Singapore; e0886363@u.nus.edu (C.E.Y.); gwendolyn.lim@u.nus.edu (G.S.J.L.); asher.ang@u.nus.edu (A.Y.H.A.); yuliang.lim@mohh.com.sg (Y.L.L.); kewinsiah@nus.edu.sg (K.T.H.S.); 2Department of Internal Medicine, Singapore General Hospital, Singapore 169608, Singapore; orlanda.goh.q.m@singhealth.com.sg; 3SingHealth Duke-NUS Medicine Academic Clinical Programme, Duke-NUS Medical School, Singapore 169857, Singapore; 4SingHealth Duke-NUS Global Health Institute, Singapore 169857, Singapore; 5Division of Gastroenterology & Hepatology, National University Hospital, Singapore 119074, Singapore; 6Saw Swee Hock School of Public Health, National University of Singapore, Singapore 117597, Singapore

**Keywords:** irritable bowel syndrome, overweight, obesity, risk factor, lifestyle, systematic review, meta-analysis

## Abstract

Background: Irritable bowel syndrome (IBS) is a common yet debilitating disorder of gut–brain interaction, characterized by gut–brain axis dysregulation, visceral hypersensitivity, and other comorbidities. Obesity has been hypothesized to be a risk factor linked to IBS, albeit evidence remains conflicting. Given the growing global prevalence of obesity and IBS, we performed a meta-analysis examining their purported association. Methods: Embase, MEDLINE, and the Cochrane Library were searched to identify studies reporting the prevalence and odds ratios (ORs) of IBS according to BMI categories. Random effects meta-analyses were used for the primary analysis. Results: From 1713 articles, 27 studies were included. Our findings showed that using study-defined categories for overweight, obese, and normal BMI, the odds of the diagnosis of IBS were not associated with overweight (OR 1.02; 95% CI 0.89 to 1.17; *p* = 0.772) or obese BMI (OR 1.11; 95% CI 0.91 to 1.37; *p* = 0.309). The meta-analysis of study-reported adjusted odds ratios of IBS among individuals living with overweight or obesity also did not yield significant results. Further sensitivity analysis by the Rome criteria demonstrated a statistically significant association between obese BMI and IBS in studies using the Rome IV criteria (OR 1.59; 95% CI 1.13 to 2.23; *p* < 0.01), with significant subgroup difference between studies using the Rome II, Rome III, and Rome IV criteria. Further sensitivity analysis using the different cut-off values and subgroup analysis by geographical territory did not yield significant associations. Conclusions: In summary, excess body weight may not be a primary driver of IBS risk. Future research should focus on longitudinal studies that account for changes in weight and other lifestyle factors, as well as detailed mechanistic investigations.

## 1. Introduction

Irritable bowel syndrome (IBS) is a prevalent and debilitating disorder of gut–brain interaction [1]. It has a global prevalence ranging from 4–8%, varying depending on the population studied, diagnostic criteria used, and geographic region [1]. It is characterized by complex pathophysiology involving gut–brain axis dysregulation, visceral hypersensitivity, gastrointestinal dysmotility, low-grade intestinal inflammation, gut dysbiosis, and psychological factors [2]. Today, IBS is one of the most frequent conditions managed in outpatient primary care and gastroenterology clinics, leading to increased healthcare utilization and costs [3].

From a preventive medicine perspective, epidemiological research has identified various risk factors associated with IBS, including gender, age, ethnicity, diet, and psychological comorbidities [4]. Obesity, particularly, has emerged as a putative risk factor linked to the development of IBS [5]. Obesity, which is commonly defined as a body mass index (BMI) of 30 kg/m^2^ or higher [6], has been associated with increased chronic low-grade inflammation, derangements in intestinal permeability, and alterations in the gut microbiota, all of which are believed to contribute to the development of gastrointestinal disorders, including IBS [5]. Although a population-based study in Japan found a statistically significant association between obesity and the prevalence of IBS (for females but not males), a birth cohort study in New Zealand did not report a statistically significant association between obesity and IBS [7].

The mechanisms connecting obesity and IBS remain speculative, though increased immunological activation and inflammation in the gut and visceral hypersensitivity have been proposed as a key factor [8]. Visceral hypersensitivity, a hallmark of IBS, involves an increased sensitivity to normal bowel function and lower pain thresholds during bowel distension [9]. It is hypothesized that obesity may exacerbate visceral hypersensitivity through chronic low-grade gut inflammation, which may also increase intestinal permeability and disturb gut microbiota balance [8].

Given the concerning trend of growing global prevalence of obesity and IBS [10], and conflicting findings on the association between the two up until now, it is thus crucial for a review to systematically explore and quantify the association between these conditions. Understanding the relationship between overweight/obesity and IBS may provide hypotheses for future research as well as valuable insights into potential interventions, such as weight management, that could alleviate IBS symptoms and improve patient outcomes. This systematic review and meta-analysis thus aimed to estimate the association between overweight/obesity and the likelihood of IBS. By synthesizing the available evidence, we seek to clarify the relationship between these conditions and identify potential areas for future research and intervention.

## 2. Methods

The study protocol was registered in PROSPERO (registration number CRD42024598790). This study was performed according to the Preferred Reporting Items for Systematic Reviews and Meta-analyses (PRISMA) and the Meta-analysis of Observational Studies (MOOSE) reporting guidelines [11,12].

### 2.1. Search Strategy

A comprehensive search of MEDLINE, Embase, and the Cochrane Library was conducted using search terms related to “irritable bowel syndrome”, “overweight”, “obesity”, and “body mass index” from 1 January 1989 (as the diagnostic criteria for IBS were first conceived in 1989 [13]) to 14 September 2024. The full search strategy for the various databases can be found in the Appendix A. The grey literature was also searched by reviewing bibliographies of included studies as well as review articles.

### 2.2. Inclusion and Exclusion Criteria

We included studies which (a) were original observational studies and (b) included data on the prevalence or odds or risk of IBS (as diagnosed with any recognized criteria such as the Rome or Manning criteria) in participants with normal, overweight, or obese BMI. Studies were excluded if they (a) were reviews, commentaries, editorials, case series/reports, abstracts, and conference proceedings, (b) did not include sufficient data to make a quantitative comparison with participants with normal BMI, (c) focused on a predominantly pediatric population or did not have a formal diagnosis of IBS, or (d) were published before the year 1989, as the guidelines for IBS were first conceived in 1989 [13].

### 2.3. Data Extraction

Two of three reviewers (C.E.Y., G.L.S.Y. or A.Y.H.A.) independently screened the titles, abstracts, and full texts of all studies in accordance with the predefined inclusion and exclusion criteria. Discrepancies were resolved after discussion with a third independent reviewer (senior author, Q.X.N.).

For the included studies after full-text review, data were extracted by two reviewers independently using a standardized data extraction form. The extracted data included study characteristics (e.g., author, year, country), sample size (of exposure and comparator group), IBS diagnosis criteria, and relevant outcomes (e.g., IBS prevalence and adjusted odds ratios (ORs) or risk ratios (RRs), along with the corresponding 95% confidence intervals).

### 2.4. Quality Assessment

For the assessment of quality and risk of bias in the included studies, two different tools were employed based on the study design. The Agency for Healthcare Research and Quality (AHRQ) checklist [14] was used to evaluate cross-sectional studies. This checklist consists of 11 items, with responses recorded as “Yes”, “No”, or “Unclear”. For each item answered “Yes”, a score of 1 was assigned, while responses marked as “No” or “Unclear” received a score of 0 (Agency of Healthcare Research and Quality, 2010). Based on the total score, studies were classified as good (8–11 points), fair (4–7 points), or poor (0–3 points).

For cohort studies and case–control studies, the Newcastle–Ottawa Scale (NOS) was used. The NOS assesses studies across three broad domains: the selection of study groups, the comparability of the groups, and the ascertainment of exposure [15]. Studies were classified as good, adequate, or poor. The assessments were conducted independently by two reviewers at the study level, with discrepancies resolved by consensus.

We evaluated the confidence in the pooled estimates using the Grading of Recommendations, Assessment, Development, and Evaluation (GRADE) approach [16]. The confidence in these estimates was classified into one of four levels: high, moderate, low, or very low.

### 2.5. Statistical Analysis

We performed all quantitative analyses in R (version 4.1.2) using the meta package. Statistical significance was determined by a two-sided *p*-value of <0.05. Using random-effects models, we performed aggregate data meta-analysis of the study-reported prevalence, OR, or RR of IBS in participants with study-defined overweight/obese BMI and compared that against that in participants with normal BMI. Heterogeneity was evaluated with I^2^ statistics, where I^2^ values below 30% were considered low, 30–60% moderate, and above 60% substantial [17].

Subgroup and sensitivity analyses by geography, BMI cut-off values, and Rome criteria were conducted. Publication bias was checked visually through funnel plot asymmetry and quantitatively using the Egger’s test [18], with a *p*-value < 0.10 considered indicative of significant bias.

## 3. Results

### 3.1. Retrieval of the Literature

A total of 1713 publications (Figure 1) were initially identified by searching MEDLINE, Embase, and Cochrane databases after removing duplicates. As shown in Figure 1, 104 articles were selected for the full-text sieve based on the title and abstract screening, of which 27 [8,19,20,21,22,23,24,25,26,27,28,29,30,31,32,33,34,35,36,37,38,39,40,41,42,43,44] studies with 543,783 participants were included in the final meta-analysis. Of these, 21 studies were cross-sectional, five studies were case–control studies, and one study was a cohort study. The risk of bias in the included studies was rated fair to good. The detailed risk of bias is displayed in Appendix A. Articles deemed ineligible during the full-text sieve are listed in Appendix A. The GRADE evidence profile was rated as low certainty and is displayed in Appendix A.

### 3.2. Characteristics of Studies Reviewed

The key summary characteristics of the included studies are outlined in Table 1. Two studies used the older Rome II criteria to diagnose IBS, while sixteen used the Rome III criteria, and nine used the Rome IV criteria. The majority of studies did not specify the specific IBS subtype for the participants, e.g., IBS with diarrhoea (IBS-D), IBS with constipation (IBS-C), and mixed IBS (IBS-M). Five studies were conducted in France, three each were conducted in Sweden, Saudi Arabia, Mexico, and Iran, two were conducted in Bangladesh, and one each was conducted in China, Egypt, Israel, Japan, Jordan, Malta, Singapore, and Turkey.

### 3.3. Meta-Analysis of Primary and Secondary Outcomes

As shown in Figure 2, using study-defined categories for overweight, obese, and normal BMI, meta-analysis of 60,306 participants with overweight BMI compared with 374,440 participants with normal BMI showed that the odds of diagnosis of IBS were insignificantly associated with overweight BMI (pooled OR 1.02; 95% CI 0.89 to 1.17; *p* = 0.772; I^2^ = 58%).

According to the meta-analysis of 33,391 participants with obese BMI compared with 374,422 participants with normal BMI, the odds of diagnosis of IBS were insignificantly associated with obese BMI, as shown in Table 2 (pooled OR 1.11; 95% CI 0.91 to 1.37; *p* = 0.309; I^2^ = 62%).

Meta-analysis of 33,237 participants with obese BMI compared with 59,753 participants with overweight BMI showed that the odds of diagnosis of IBS were insignificantly associated with obese BMI, as shown in Table 2 (pooled OR 1.04; 95% CI 0.94 to 1.15; *p* = 0.459; I^2^ = 14%).

Meta-analysis of 93,697 participants with normal BMI compared with 375,101 participants with overweight or obese BMI showed that the odds of diagnosis of IBS were insignificantly associated with higher BMI, as shown in Table 2 (pooled OR 1.09; 95% CI 0.94 to 1.26; *p* = 0.272; I^2^ = 69%).

In the meta-analysis of study-reported adjusted odds ratios of IBS in participants with overweight BMI compared to participants with normal BMI (forest plot shown in Figure 3), overweight BMI was insignificantly associated with IBS diagnosis (OR 0.94; 95% CI 0.87 to 1.02; *p* = 0.166; I^2^ = 50%).

In the meta-analysis of study-reported adjusted odds ratios of IBS in participants with obese BMI compared to participants with normal BMI (forest plot shown in Figure 4), obese BMI was insignificantly associated with IBS diagnosis (OR 0.90; 95% CI 0.77 to 1.06; *p* = 0.218; I^2^ = 30%).

### 3.4. Sensitivity and Subgroup Analyses

Further sensitivity analysis by the Rome criteria demonstrated a statistically significant association between obese BMI and IBS in studies using the Rome IV criteria (OR 1.59; 95% CI 1.13 to 2.23; *p* < 0.01; I^2^ = 49%) (as shown in Table 3), with significant subgroup difference between studies using the Rome II, Rome III, and Rome IV criteria.

We attempted a further sensitivity analysis (Table 3) using studies which defined normal BMI as 18.5 kg/m^2^ to 24.9 kg/m^2^, but this yielded no significant results. The limited number of studies (n = 2) using the World Health Organization’s (WHO) Asian BMI cut-off values [45] precluded meaningful analysis.

Heterogeneity between studies was substantial (I^2^ = 69%) for the overall analysis, but was reduced in certain subgroups. Subgroup analysis by continental regions delineated by the United Nations demonstrated no continent where there is a statistically significant association between overweight or obese BMI and IBS diagnosis (Table 3). Unfortunately, subgroup analysis by the specific IBS subtype (e.g., IBS-C, IBS-D, or IBS-M) was not possible due to the lack of data.

A sensitivity analysis removing two studies ([26,28]) which used BMI < 30 kg/m^2^ as the normal BMI cut-off showed no significant association between obese BMI and IBS (OR 1.14; 95% CI 0.86 to 1.52; I^2^ = 61%).

### 3.5. Assessment of Publication Bias

Assessment of funnel plots (displayed in the Appendix A) and Egger’s test did not show significant publication bias for the analysis of individuals with overweight BMI (*p* = 0.954) and obese BMI (*p* = 0.587).

## 4. Discussion

This systematic review and meta-analysis aimed to investigate the association between overweight/obesity and IBS. Despite the biologically plausible links between obesity and IBS, including shared pathophysiological mechanisms such as chronic low-grade inflammation, altered gut microbiota, and visceral hypersensitivity, our findings mostly did not find a significant association between higher BMI and the risk of developing IBS. We did, however, demonstrate a significant association between obese BMI and IBS in studies using the Rome IV criteria. These results suggest that while obesity may exacerbate gastrointestinal symptoms in some individuals, it may not be a strong independent risk factor for IBS.

Our findings align with some previous studies that have reported inconsistent or nonsignificant associations between obesity and functional gastrointestinal disorders (FGIDs), including IBS. For instance, large population-based studies in Europe and Asia [21,30,40,46] have found mixed-to-negative results, with some indicating no significant association between BMI and IBS, while others reported that factors such as abdominal adiposity rather than BMI alone may play a more critical role in the development of FGIDs. The lack of a significant association in our analysis could be due to several reasons. First, IBS is a multifactorial disorder influenced by a complex interplay of genetic, environmental, and psychosocial factors [47]. While obesity contributes to some of these factors, such as chronic inflammation and altered gut microbiota, it may not be sufficient to trigger IBS on its own. Second, the heterogeneity in IBS diagnostic criteria, particularly the transition from Rome III to Rome IV criteria, may contribute to the variability in study outcomes, making it difficult to detect a consistent relationship between BMI and IBS across different populations. Third, visceral fat is thought to be more closely associated with inflammatory and metabolic processes than subcutaneous fat [48,49], and BMI alone may be inadequate for thoroughly assessing the relationship between overweight/obesity and IBS risk, as reported by an earlier study by Lee et al. [50]. The levels of circulating interleukin (IL)-6 have frequently been reported to be higher among IBS patients than healthy controls, indicating a pro-inflammatory phenotype in these patients. Rather than BMI alone, visceral adiposity has been found to be strongly associated with IL-6 levels [51]. This could thus explain the lack of significant association found in our analysis. Fourth, the morbidity and functional limitation among those living with obesity may vary with class and we would expect most conditions to be more prevalent and/or severe with higher classes of obesity. It would be worthwhile seeing what the associations are if we were to analyze the data by class I, II, and III obesity, as per WHO guidelines [45].

We did, however, observe in our sensitivity analysis that in studies which used the Rome IV criteria for IBS diagnosis, participants with obese BMI were more likely to be diagnosed with IBS, with a significant subgroup difference among studies which used the Rome II, Rome III, and Rome IV criteria. The current literature suggests that the Rome IV IBS population likely reflects a subgroup of patients who would be diagnosed as having IBS using the Rome III criteria; in the Rome IV IBS population, patients are more likely to have more severe GI symptoms (as the criteria require abdominal pain rather than discomfort as a qualifying symptom), and lower quality of life [52,53]. While the association between the increased symptomatology observed in Rome IV IBS population and higher BMI remains unclear, studies have hypothesized that obesity, with its associated low-grade chronic inflammation, can lead to increased serum levels of inflammatory molecules such as complement components and C-reactive protein, further exacerbating IBS symptoms [8]. However, there are likely to be potential differences particularly regarding symptom severity and potential comorbidities, such as obesity-related inflammation. The change in terminology used in the Rome IV criteria from functional gastrointestinal disorders to disorders of gut–brain interaction [54] also emphasizes our evolving understanding of the complex interaction between the gut and the nervous system. Further studies should investigate the mechanistic pathways mediating the relationship between obesity and IBS symptomology and severity, using a consistent set of diagnostic criteria.

Although our meta-analysis did not find a significant association between obesity and IBS, it is essential to consider the potential mechanisms through which obesity may influence IBS symptoms in specific subgroups. Visceral hypersensitivity, for example, remains a well-established pathophysiological mechanism in IBS. Obesity may still play a role in exacerbating symptoms through increased visceral fat and low-grade inflammation, potentially contributing to the hypersensitivity of visceral afferent pathways. Abdominal obesity is also known to increase intra-abdominal pressure, which can worsen symptoms like bloating, constipation, and abdominal pain [55], all common in IBS. However, these effects may be more pronounced in individuals with abdominal obesity or metabolic syndrome [56], rather than in the broader population of individuals with elevated BMI alone. Moreover, obesity and IBS share common psychosocial comorbidities, including anxiety, depression, and stress, which are known to affect the gut–brain axis.

Additionally, the lack of a significant association in our findings may indicate that the relationship between obesity and IBS is more indirect, mediated through other factors (including psychiatric comorbidities) rather than a direct consequence of excess body weight. Some psychosocial factors including stress, anxiety and depression are common in both individuals living with obesity and IBS [43,57,58]. Chronic stress could trigger cortisol release, which can disrupt normal GI motility and alter gut microbiota [59,60], potentially inciting IBS symptoms and influencing adverse eating behaviors. Similarly, dysregulation in the gut–brain signaling could lead to changes in hunger and satiety cues [61,62], leading to changes in eating behaviors and impacting both IBS and obesity. The population living with obesity is quite heterogeneous; a subgroup with a stronger history of obesogenic diets, which includes a predominance of processed foods, may be more predisposed to IBS. This has implication for future research as we work to better identify this at-risk group and provide more tailored nutritional intervention for them.

### Limitations of Review

Despite employing a comprehensive search strategy and rigorous methodology, several limitations should be acknowledged in this systematic review and meta-analysis. Firstly, there was still substantial heterogeneity across the included studies, which may have diluted any potential associations between overweight/obesity and IBS. This heterogeneity stemmed from differences in study design, populations, geographic regions, and diagnostic criteria for both IBS and obesity. Such variability can complicate the interpretation of pooled estimates and reduce the precision of our findings. We attempted to identify sources of heterogeneity and perform the relevant subgroup and sensitivity analyses, which reduced the level of heterogeneity in the estimates. The relatively high residual heterogeneity in the sensitivity analysis of BMI definitions could be due to the high OR reported by the study by Mohammed et al. [43]. This was the only study conducted in Africa, and its setting was in a rural family practice center, markedly different from the remaining studies. Thus, complex biological and psychosocial factors could have influenced the prevalence of IBS in this unique setting. Future research should be directed towards investigating IBS incidence in different geographical regions and settings. Related to this, ideally, a subgroup analysis should have been performed for the different IBS subtypes. However, most of the studies did not specify the IBS subtype under investigation, precluding the possibility of a detailed subgroup exploration. Future research should investigate the differential influence that obesity and other indices of adiposity have on IBS symptomology. The existing literature indicates that obesity-related impacts may vary between constipation-predominant (IBS-C) and non-constipation subtypes (IBS-D and IBS-M), given the influence of body mass on bowel habits and transit times [63]. Moreover, other studies have suggested that IBS-D and IBS-C could have different pathways by which inflammation arises. In a case–control study by Lee et al. [50], they showed that three different measures of visceral adiposity were associated with IBS-D, but not IBS-C.

Secondly, there was notable inconsistency in the definitions of overweight and obesity across the studies. While the WHO has recommended specific BMI cut-offs for defining overweight and obesity in different ethnic groups—such as lower thresholds for Asian populations—many of the included studies used diverse BMI cut-offs that were not in accordance with these recommendations [26,28,33]. This variability in the classification of BMI could have introduced residual confounding, as it does not account for population-specific differences in body composition and metabolic risk. Although we attempted to address this through sensitivity analyses using WHO-defined BMI cut-offs, these analyses did not yield significant associations, likely due to the small number of studies adhering to these specific thresholds. This also means that we should interpret with caution for the studies involving the non-White populations.

Thirdly, the observational nature of the included studies limits our ability to draw definitive conclusions about causality. While we observed associations between BMI and IBS in certain subgroups, it is important to recognize that these studies are susceptible to bias, including residual confounding and reverse causation. The relationship between obesity and IBS is likely bidirectional, meaning that while obesity may exacerbate IBS symptoms through mechanisms such as altered gut motility and increased inflammation, IBS itself could influence dietary habits, leading to weight changes. It is not uncommon for IBS patients to modify their diet by avoiding specific items they perceive as problematic, and in some cases, these patients may eliminate entire food groups [64]. It is admittedly difficult to determine the direction of causality from the available cross-sectional data.

Lastly, our reliance on BMI as a single measure of obesity presents its own limitations. BMI, while widely used, is a simple anthropometric measure of body mass that does not differentiate between fat and lean tissue or capture fat distribution, particularly visceral adiposity, which is more strongly associated with metabolic and inflammatory processes [48,49]. Visceral fat, in particular, has been shown to have a greater impact on gastrointestinal health and inflammation than BMI alone can reveal. Other studies report significant associations between IBS and alternative measures for obesity. A study comparing patients with IBS and healthy controls found that fat free mass measured via bioelectrical impedance analysis was lower in patients with IBS and that this was correlated with symptom severity [65]. Lee et al. [50] demonstrated significantly increased risk of IBS in patients with increased visceral adipose tissue area, visceral adipose to subcutaneous adipose tissue ratio and waist circumference. These findings underscore the value of evaluating the relationship between IBS and more diverse metrics of abdominal obesity. Future studies would benefit from incorporating more precise measures of body fat distribution, such as waist-to-hip ratio or imaging-based assessments, to better understand the role of visceral adiposity in IBS.

## 5. Conclusions

In conclusion, excess body weight may not be a primary driver of IBS risk given the non-significant findings based on currently available observational evidence. Future research should focus on more longitudinal studies that account for changes in weight and other lifestyle factors over time, as well as detailed mechanistic investigations, which would provide greater insight into the relationship between overweight/obesity and IBS.

## Figures and Tables

**Figure 1 nutrients-16-03984-f001:**
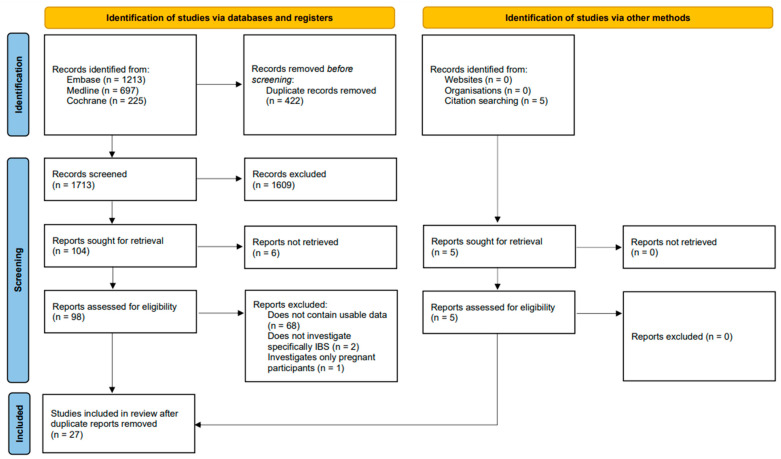
PRISMA flowchart showing the study selection process.

**Figure 2 nutrients-16-03984-f002:**
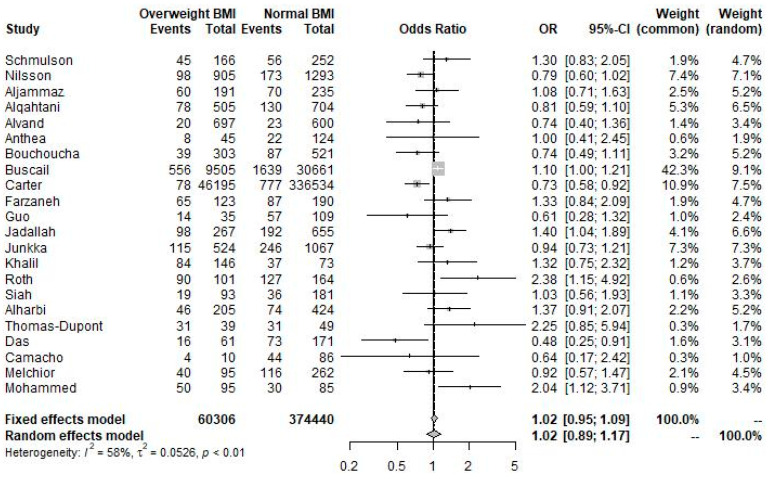
Forest plot showing the pooled odds ratio of IBS in participants with overweight BMI versus participants with normal BMI using study-reported prevalence data.

**Figure 3 nutrients-16-03984-f003:**
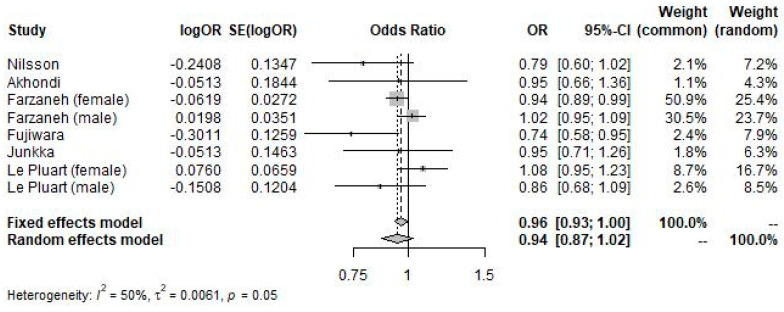
Forest plot showing the pooled odds ratio of IBS in participants with overweight BMI versus participants with normal BMI using study-reported adjusted odds ratios.

**Figure 4 nutrients-16-03984-f004:**
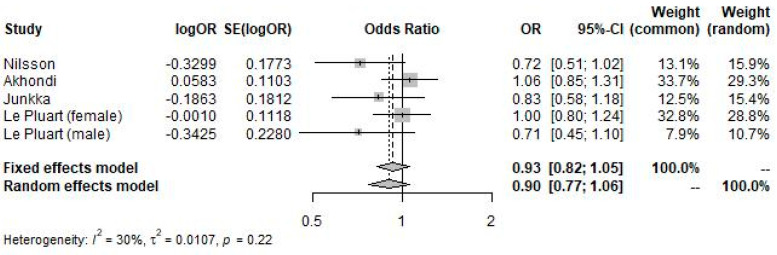
Forest plot showing the pooled odds ratio of IBS in participants with obese BMI versus participants with normal BMI using study-reported adjusted odds ratios.

**Table 1 nutrients-16-03984-t001:** Summary of study characteristics and findings.

Author, Year	Study Type	IBS Diagnosis Criteria	Country of Study	Definition of Overweight BMI (kg/m^2^)	Definition of Obese BMI (kg/m^2^)	Definition of Normal BMI (kg/m^2^)	Total Number of Participants	Number of Participants with IBS	Number of Participants Without IBS
Akhondi, 2019 [21]	Cross-sectional	Rome III	Iran	25–29.9	≥30	<25	4763	NR	NR
Aljammaz, 2020 [22]	Cross-sectional	Rome III	Saudi Arabia	>25	NR	≤25	426	130	296
Alqahtani, 2022 [23]	Cross-sectional	Rome IV	Saudi Arabia	NR	NR	NR	1680	306	1374
Alvand, 2020 [24]	Cross-sectional	Rome III	Iran	NR	NR	NR	1849	60	1789
Anthea, 2021 [25]	Cross-sectional	Rome IV	Malta	25–30	>30	18.5–25	192	34	158
Bayrak, 2020 [26]	Case–control	Rome IV	Turkey	NR	≥30	<30	614	310	304
Bouchoucha, 2016 [27]	Cross-sectional	Rome III	France	25–30	≥30	18.5–25	1074	150	924
Brochard, 2017 [28]	Case–control	Rome III	France	NR	≥30	<30	201	106	95
Buscail, 2017 [29]	Cross-sectional	Rome III	France	25–30	≥30	<25	44,350	2423	41,927
Carter, 2015 [30]	Cohort	Rome II	Israel	NR	NR	NR	440,822	976	438,846
Farzaneh, 2013 [31]	Case–control	Rome III	Iran	≥25.0	NR	<25	316	153	163
Fujiwara, 2011 [32]	Cross-sectional	Rome III	Japan	>25.0	NR	≤25.0	2680	381	2299
Guo, 2015 [33]	Case–control	Rome III	China	24.0–27.9	≥28.0	18.5–23.9	157	78	79
Jadallah, 2022 [34]	Cross-sectional	Rome III	Jordan	NR	NR	NR	1094	338	756
Junkka, 2023 [35]	Cross-sectional	Rome III	Sweden	25.0–29.9	>30.0	<25.0	1915	436	1479
Khalil, 2024 [36]	Cross-sectional	Rome IV	Bangladesh	>23.0	NR	≤23.0	219	121	98
Le Pluart, 2015 [37]	Cross-sectional (female)	Rome III	France	25.0–30.0	>30.0	18.5–25	27,617	1539	26,078
	Cross-sectional (male)	Rome III	France	25.0–30.0	>30.0	<25.0	7830	358	7472
Roth, 2024 [38]	Cross-sectional	Rome IV	Sweden	≥25.0	≥30.0	<25	310	260	50
Siah, 2016 [39]	Cross-sectional	Rome III	Singapore	≥25.0	NR	18.5–25	297	62	235
Thomas-Dupont, 2022 [8]	Cross-sectoinal	Rome IV	Mexico	25.0–30.0	>30.0	18.5–25	114	79	35
Das, 2022 [40]	Cross-sectional	Rome III	Bangladesh	23–27.4	≥27.5	18.5–22.9	300	118	182
Camacho, 2023 [41]	Cross-sectional	Rome IV	Mexico	25.0–29.9	≥30.0	18.0–24.9	105	51	54
Melchior, 2020 [42]	Case–control	Rome III	France	25.0–29.9	≥30	18.5–25	456	228	228
Mohammed, 2021 [43]	Cross-sectional	Rome IV	Egypt	25.0–29.9	≥30	18.5–25	350	175	175
Alharbi, 2022 [44]	Cross-sectional	Rome IV	Saudi Arabia	25–29.9	>30	18.5–25	921	186	735
Schmulson, 2010 [19]	Cross-sectional	Rome II	Mexico	≥25	≥30	18.5–25	483	113	370
Nilsson, 2021 [20]	Cross-sectional	Rome III	Sweden	25–29.9	≥30	<25	2648	316	2332

Abbreviation: NR, Not Reported.

**Table 2 nutrients-16-03984-t002:** Summary of pooled ORs of IBS among overweight and obese individuals.

Comparison Group	Control Group	Number of Studies	Number of Patients in Comparison Group	Number of Patients in Control Group	Pooled OR (95% CI) Based on Random-Effects Model	I^2^ Statistic
Overweight BMI	Normal	22	60,306	374,440	1.02 (0.89, 1.17)	58%
Obese BMI	Normal	20	33,391	374,422	1.11 (0.91, 1.37)	62%
Obese BMI	Overweight	18	33,237	59,753	1.04 (0.94, 1.15)	14%
Overweight BMI and Obese BMI	Normal	24	93,697	375,101	1.09 (0.94, 1.26)	69%

**Table 3 nutrients-16-03984-t003:** Findings of sensitivity and subgroup analyses.

Variable	Comparison Group	Control Group	Number of Studies	Number of Patients in Comparison Group	Number of Patients in Control Group	Pooled OR (95% CI) Based on Random-Effects	I^2^	Test for Subgroup Differences, Random-Effects
Rome criteria								
Rome II	Obese BMI	Normal BMI	2	27,137	336,786	0.68 (0.51, 0.90)	0%	<0.01
Rome III			10	5440	35,473	1.00 (0.81, 1.22)	47%	
Rome IV			8	814	2163	1.59 (1.13, 2.23)	49%	
Rome II	Overweight BMI	Normal BMI	2	46,361	336,786	0.94 (0.54, 1.65)	80%	0.17
Rome III			12	12,799	35,945	0.95 (0.82, 1.11)	51%	
Rome IV			8	1146	1709	1.33 (0.97, 1.83)	56%	
Continent								
North America	Obese BMI	Normal BMI	3	93	387	0.88 (0.51, 1.54)	0%	0.82
Europe			8	4820	34,226	1.06 (0.78, 1.46)	66%	
Asia			8	28,338	339,724	1.07 (0.81, 1.42)	59%	
North America	Overweight BMI	Normal BMI	3	215	387	1.34 (0.90, 1.98)	13%	0.31
Europe			7	11,478	34,092	0.97 (0.82, 1.15)	58%	
Asia			11	48,518	339,876	0.97 (0.79, 1.19)	61%	
Studies using 18.5 kg/m^2^ to 24.9 kg/m^2^ as normal BMI								
As above	Obese BMI	Normal BMI	7	637	1717	1.20 (0.75, 1.92)	72%	NA
As above	Overweight BMI	Normal BMI	8	1041	1898	1.17 (0.91, 1.52)	41%	NA

## Data Availability

The data that support the findings of review are derived from publicly available sources. All included studies and datasets were obtained from published research articles, which are accessible via databases such as Embase, MEDLINE, and the Cochrane Library. No new primary data were generated in the course of this research.

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
