# Peer review of "Examining the Association Between Overweight, Obesity, and Irritable Bowel Syndrome: A Systematic Review and Meta-Analysis"

_nutrients, 2024, doi:10.3390/nu16233984_

Round 1

Reviewer 1 Report

Comments and Suggestions for Authors

Comments to authors

Introduction

·        “This systematic review and meta-analysis thus aimed to investigate the association be-69 tween overweight/obesity and the likelihood of IBS.”: Please change investigate to estimate.

Methods

·        Please include subsections (search strategy, inclusion/exclusion criteria, risk of bias assessment, data extraction, etc.).

·        First paragraph: as your systematic review is of observational studies, you must include and cite the MOOSE guide (in addition to PRISMA).

·        Lines 77-83: AMSTAR-2 requires a grey literature search. Please indicate that you have searched for grey literature (and have actually searched). This means that regardless of whether you find results in grey literature, you must search in grey literature.

·        Include a subsection on "Grading the quality of evidence" using the GRADE tool in the Methods (and therefore in the Results).

·        Typically, the risk of bias or quality of studies sections are placed before the statistical analyses section.

·        In the "Statistical analyses" section, some things are not cited: classification of heterogeneity, Egger test, etc.

Results

·        In the supplementary material, please include a table with the studies that were read in full and excluded. That is, include the list of studies that were excluded after reading the full text and their main reason for exclusion.

·        In this section, “showed that the odds of diagnosis of IBS were insignificantly 167 raised in participants with obese BMI” or “the odds of diagnosis of IBS were insignificantly 160 higher among obese individuals, as shown in Table 2” or…. should be avoided. I mean, if the confidence interval includes 1, there is simply no association, we could only say (if we wanted to) that there is a trend... But in any case, if something is not statistically significant, there is no association, it neither increases nor decreases the risk.

Discussion

Ok

References

Ok

Author Response

Reviewer 1:

Introduction

“This systematic review and meta-analysis thus aimed to investigate the association between overweight/obesity and the likelihood of IBS.”: Please change investigate to estimate.

Our Response:

We thank the reviewer for this suggestion. We have now changed “investigate” to “estimate”.

Methods

Please include subsections (search strategy, inclusion/exclusion criteria, risk of bias assessment, data extraction, etc.).

Our Response:

We thank the reviewer for this suggestion. We have now added the relevant subsections.

First paragraph: as your systematic review is of observational studies, you must include and cite the MOOSE guide (in addition to PRISMA).

Our Response:

We thank the reviewer for this suggestion. We have now included and cited reporting guidelines of MOOSE. We have now attached the completed MOOSE checklist as a supplementary document.

Lines 77-83: AMSTAR-2 requires a grey literature search. Please indicate that you have searched for grey literature (and have actually searched). This means that regardless of whether you find results in grey literature, you must search in grey literature.

Our Response:

We have now further clarified in the main text and in our Figure 1 that we did search the grey literature and found relevant articles to supplement our systematic search:

“Grey literature was also searched by reviewing bibliographies of included studies as well as review articles.”

Include a subsection on "Grading the quality of evidence" using the GRADE tool in the Methods (and therefore in the Results).

Our Response:

We have now included the GRADE tool methodology and results in the manuscript:

“We evaluated the confidence in the pooled estimates using the Grading of Recommendations, Assessment, Development, and Evaluation (GRADE) approach (16). The confidence in these estimates was classified into one of four levels: high, moderate, low, or very low.”

“The GRADE evidence profile was rated as low certainty, and is displayed in Supplementary Table S4.”

Typically, the risk of bias or quality of studies sections are placed before the statistical analyses section.

In the "Statistical analyses" section, some things are not cited: classification of heterogeneity, Egger test, etc.

Our Response:

We have now cited the relevant sources for the aforementioned statistical tests:

“We performed all quantitative analyses in R (version 4.1.2) using the meta package. Statistical significance was determined by a two-sided P value of <0.05. Using ran-dom-effects models, we performed aggregate data meta-analysis of the study-reported prevalence or OR or RR of IBS in participants with study-defined overweight/obese BMI and compared that against that in participants with normal BMI. Heterogeneity was evaluated with I2 statistics, where I² values below 30% were considered low, 30%-60% moderate, and above 60% substantial (17).

Subgroup and sensitivity analyses by geography, BMI cut-off values and Rome criteria were conducted. Publication bias was checked visually through funnel plot asymmetry and quantitatively using the Egger’s test (18), with a p-value <0.10 consid-ered indicative of significant bias.”

Results

In the supplementary material, please include a table with the studies that were read in full and excluded. That is, include the list of studies that were excluded after reading the full text and their main reason for exclusion.

Our Response:

We have now included the list of studies that were excluded in the full text sieve in Supplementary Table S3:
“Articles deemed ineligible during the full-text sieve are listed in Supplementary Table S3.”

In this section, “showed that the odds of diagnosis of IBS were insignificantly 167 raised in participants with obese BMI” or “the odds of diagnosis of IBS were insignificantly 160 higher among obese individuals, as shown in Table 2” or…. should be avoided. I mean, if the confidence interval includes 1, there is simply no association, we could only say (if we wanted to) that there is a trend... But in any case, if something is not statistically significant, there is no association, it neither increases nor decreases the risk.

Our Response:

We have now reworded the text in the manuscript and the abstract to reflect the lack of association between BMI and IBS.

Reviewer 2 Report

Comments and Suggestions for Authors

This meta-analysis provides a thorough investigation into the association between overweight/obesity and irritable bowel syndrome (IBS), incorporating data from 27 studies and using rigorous methods to address this important research question. By exploring how BMI categories relate to IBS diagnosis and analyzing the role of different Rome criteria, your work sheds light on the complex interplay between these conditions. However, to further strengthen your paper and ensure clearer, more impactful conclusions, I recommend the following revisions:

  1. Address Study Heterogeneity: The substantial heterogeneity observed (I² values up to 69%) weakens the robustness of your findings. Consider performing a meta-regression or additional refined subgroup analyses to explore potential sources of variability and explain them more clearly in your discussion.

  2. Clarify BMI Limitations: Your reliance on BMI as a measure of obesity is a noted limitation. Highlight this more explicitly and discuss how using alternative metrics, like visceral fat assessment or waist-to-hip ratio, could yield more accurate insights. Encourage future research to incorporate these measures.

  3. Expand on Rome Criteria Differences: Since IBS diagnosis can vary significantly with different Rome criteria (II, III, IV), provide a more detailed discussion of how these differences might influence your findings and the broader implications for interpreting IBS research.

  4. Acknowledge the Lack of IBS Subtype Data: The inability to analyze IBS subtypes (e.g., IBS-C, IBS-D) is a key limitation. Discuss how this may have affected your results and emphasize the importance of future studies focusing on subtype-specific analyses.

  5. Enhance the Mechanistic Discussion: While your discussion of inflammation, visceral hypersensitivity, and gut microbiota is insightful, consider briefly elaborating on additional psychosocial or gut-brain axis factors that may mediate the obesity-IBS relationship.

These adjustments will make your manuscript more comprehensive and address critical limitations in the current literature

Author Response

Reviewer 2:

This meta-analysis provides a thorough investigation into the association between overweight/obesity and irritable bowel syndrome (IBS), incorporating data from 27 studies and using rigorous methods to address this important research question. By exploring how BMI categories relate to IBS diagnosis and analyzing the role of different Rome criteria, your work sheds light on the complex interplay between these conditions. However, to further strengthen your paper and ensure clearer, more impactful conclusions, I recommend the following revisions:

Address Study Heterogeneity: The substantial heterogeneity observed (I² values up to 69%) weakens the robustness of your findings. Consider performing a meta-regression or additional refined subgroup analyses to explore potential sources of variability and explain them more clearly in your discussion.

Our Response:

We thank the reviewer for the invaluable suggestion. We have performed sensitivity analyses according to Rome criteria and BMI cut-off values, and subgroup analysis by geographical region to examine the reasons for the relatively substantial heterogeneity. This reduced the heterogeneity in some of the estimates. For the other estimates, we have further discussed in the discussion the possible reasons for the residual heterogeneity:

“Despite employing a comprehensive search strategy and rigorous methodology, several limitations should be acknowledged in this systematic review and meta-analysis. Firstly, there was still substantial heterogeneity across the included studies, which may have diluted any potential associations between overweight/obesity and IBS. This heterogeneity stemmed from differences in study design, populations, geographic regions, and diagnostic criteria for both IBS and obesity. Such variability can complicate the interpretation of pooled estimates and reduce the precision of our findings. We at-tempted to identify sources of heterogeneity and perform the relevant subgroup and sensitivity analyses, which reduced the level of heterogeneity in the estimates. The relatively high residual heterogeneity in the sensitivity analysis of BMI definitions could be due to the high OR reported by the study by Mohammed et al. (43) This was the only study conducted in the Africa, and its setting was in a rural family practice centre, markedly different from the remaining studies. Thus, complex biological and psycho-social factors could have influenced the prevalence of IBS in this unique setting. Future research should be directed towards investigating IBS incidence in different geograph-ical regions and settings. Related to this, ideally, a subgroup analysis should have been performed for the different IBS subtypes. However, most of the studies did not specify the IBS subtype under investigation, precluding the possibility of a detailed subgroup exploration. Future research should investigate the differential influence that obesity and other indices of adiposity have on IBS symptomology. Existing literature indicates that obesity-related impacts may vary between constipation-predominant (IBS-C) and non-constipation subtypes (IBS-D and IBS-M), given the influence of body mass on bowel habits and transit times (52). Moreover, other studies have suggested that IBS-D and IBS-C could have different pathways by which inflammation arises. In a case-control study by Lee et al. (49), they showed that three different measures of vis-ceral adiposity were associated with IBS-D, but not IBS-C.”

Clarify BMI Limitations: Your reliance on BMI as a measure of obesity is a noted limitation. Highlight this more explicitly and discuss how using alternative metrics, like visceral fat assessment or waist-to-hip ratio, could yield more accurate insights. Encourage future research to incorporate these measures.

Our Response:

We thank the reviewer for this comment. We have now emphasised the importance of future research in utilising other measures of obesity, especially ones that offer a more accurate measure of visceral adiposity, which has been linked to multiple adverse health outcomes:

“Lastly, our reliance on BMI as a single measure of obesity presents its own limitations. BMI, while widely used, is a simple anthropometric measure of body mass that does not differentiate between fat and lean tissue or capture fat distribution, particularly visceral adiposity, which is more strongly associated with metabolic and inflammatory processes (46). Visceral fat, in particular, has been shown to have a greater impact on gas-trointestinal health and inflammation than BMI alone can reveal. Other studies report significant associations between IBS and alternative measures for obesity.  A study (55) found that fat free mass measured via bioelectrical impedance analysis was lower in IBS patients and correlated to symptom severity. Lee et al. (49) demonstrated significantly increased risk of IBS in patients with increased visceral adipose tissue area, visceral adipose to subcutaneous adipose tissue ratio and waist circumference. These findings underscore the value of evaluating the relationship between IBS and more diverse metrics of abdominal obesity. Future studies would benefit from incorporating more precise measures of body fat distribution, such as waist-to-hip ratio or imaging-based assessments, to better understand the role of visceral adiposity in IBS.”

Expand on Rome Criteria Differences: Since IBS diagnosis can vary significantly with different Rome criteria (II, III, IV), provide a more detailed discussion of how these differences might influence your findings and the broader implications for interpreting IBS research.

Our Response:

We thank the reviewer for this comment. We have now expanded on the differences in Rome criteria, and how this might affect the findings in our study, along with the broader implications for future IBS research:

“Current literature suggests that the Rome IV IBS population likely reflects a subgroup of patients who would be diagnosed as having IBS using the Rome III criteria; in the Rome IV IBS population, patients are more likely to have more severe GI symptoms, and lower quality of life (51, 52). While the association between the increased symptomatology observed in Rome IV IBS population and higher BMI remains unclear, studies have hypothesised that obesity, with its associated low-grade chronic inflammation, can lead to increased serum levels of inflammatory molecules such as complement components and C-reactive protein, further exacerbating IBS symptoms (8). However, there are likely to be potential differences particularly regarding symptom severity and potential comorbidities, such as obesity-related inflammation. The change in terminology used in the Rome IV criteria from functional gastrointestinal disorders to disorders of gut-brain interaction (53) also emphasizes our evolving understanding of the complex interaction between the gut and the nervous system. Further studies should investigate the mechanistic pathways mediating the relationship between obesity and IBS symptomology and severity, using a consistent set of diagnostic criteria.”

Acknowledge the Lack of IBS Subtype Data: The inability to analyze IBS subtypes (e.g., IBS-C, IBS-D) is a key limitation. Discuss how this may have affected your results and emphasize the importance of future studies focusing on subtype-specific analyses.

Our Response:

We thank the reviewer for this comment. We have now enhanced the discussion points on how the results might have differed based on IBS subtypes, and how future research should focus on subtype-specific analyses, in order to better elucidate the mechanisms between obesity and IBS:

“Related to this, ideally, a subgroup analysis should have been performed for the different IBS subtypes. However, most of the studies did not specify the IBS subtype under investigation, precluding the possibility of a detailed subgroup exploration. Future research should investigate the differential influence that obesity and other indices of adiposity have on IBS symptomatology. Existing literature indicates that obesity-related impacts may vary between constipation-predominant (IBS-C) and non-constipation subtypes (IBS-D and IBS-M), given the influence of body mass on bowel habits and transit times (52). Moreover, other studies have suggested that IBS-D and IBS-C could have different pathways by which inflammation arises. In a case-control study by Lee et al. (49), they showed that three different measures of visceral adiposity were associated with IBS-D, but not IBS-C.”

Enhance the Mechanistic Discussion: While your discussion of inflammation, visceral hypersensitivity, and gut microbiota is insightful, consider briefly elaborating on additional psychosocial or gut-brain axis factors that may mediate the obesity-IBS relationship.

Our Response:

We thank the reviewer for this comment. We have now elaborated on additional psychosocial or gut-brain axis factors that may mediate the obesity-IBS relationship:

“The lack of a significant association in our findings may indicate that the relationship between obesity and IBS is more indirect, mediated through other factors (including psychiatric comorbidities) rather than a direct consequence of excess body weight. Some psychosocial factors include stress, anxiety and depression, which are both common in obesity and IBS (43, 54, 55). Chronic stress could trigger cortisol release which can disrupt GI motility and alter gut microbiota (56, 57), potentially worsening IBS symptoms and influencing adverse eating behaviors. Additionally, dysregulation in the gut-brain signaling could lead to changes in hunger and satiety cues (58, 59), leading to changes in eating behaviors and impacting both IBS and obesity.”

Round 2

Reviewer 1 Report

Comments and Suggestions for Authors

-

Reviewer 2 Report

Comments and Suggestions for Authors

I read this revised version of the really relevant and interesting topic of obesity and IBS- connection. Since the co-exsitence is really enormous, the results of this study are also important.

The authors replied in a satisfactory manner the reviewers' comments and I do not have any objection endorsing publication for this nice study, which will very likely contribute to the evidence linking obesity-metabolic syndrome and irritable bowel syndrome.